# Cellular structure of dinosaur scales reveals retention of reptile-type skin during the evolutionary transition to feathers

Zixiao Yang [1,2] ✉, Baoyu Jiang [3], Jiaxin Xu[3] & Maria E. McNamara [1,2]

Fossil feathers have transformed our understanding of integumentary evolution in vertebrates. The evolution of feathers is associated with novel skin ultrastructures, but the fossil record of these changes is poor and thus the critical transition from scaled to feathered skin is poorly understood. Here we shed light on this issue using preserved skin in the non-avian feathered dinosaur *Psittacosaurus*. Skin in the non-feathered, scaled torso is three-dimensionally replicated in silica and preserves epidermal layers, corneocytes and melanosomes. The morphology of the preserved stratum corneum is consistent with an original composition rich in corneous beta proteins, rather than (alpha-) keratins as in the feathered skin of birds. The stratum corneum is relatively thin in the ventral torso compared to extant quadrupedal reptiles, reflecting a reduced demand for mechanical protection in an elevated bipedal stance. The distribution of the melanosomes in the fossil skin is consistent with melanin-based colouration in extant crocodilians. Collectively, the fossil evidence supports partitioning of skin development in *Psittacosaurus*: a reptile-type condition in non-feathered regions and an avian-like condition in feathered regions. Retention of reptile-type skin in non-feathered regions would have ensured essential skin functions during the early, experimental stages of feather evolution.

Avian feathers fulfil many important roles, most importantly in flight, swimming, insulation, display, sensory function and protection against parasites[1]. As a result, feathers have long been regarded as the key innovation responsible for the emergence of flight capability in, and adaptive radiations of, birds[1,2]. Fossilised feathers and feather-like filamentous integumentary structures, however, have been reported in other groups, including non-avian theropods, ornithischians and pterosaurs[2–4]. Subsequently the term feather has been equivocal in the literature, referring to (1) structures comparable to modern feathers (e.g. ref. [5]), (2) the filamentous and pennaceous integumentary structures of theropod dinosaurs (e.g. ref. [6]), (3) the pennaceous integumentary structures of avemetatarsalians (e.g. ref. [7]) and (4) the

filamentous and pennaceous integumentary structures of avemetatarsalians (e.g. ref. [4]). Here, we follow the last approach, which defines feather in its broadest sense.

Despite decades of research, the evolutionary origins of feathers remain poorly resolved and, as a result, are the subject of ongoing debate. Certain studies favour independent evolution of feathers in theropods, ornithischians and pterosaurs[8,9], but given the shared morphology and histology of the feather structures, and the likely shared genomic heritage and shared pattern of developmental stages of these organisms, we consider a single point of origin more likely[2–4,10]. At the very least, the available fossil and developmental evidence strongly suggests that very similar, if not identical, genetic

[1]School of Biological, Earth and Environmental Sciences, University College Cork, Cork, Ireland. [2]Environmental Research Institute, University College Cork, Cork, Ireland. [3]State Key Laboratory for Mineral Deposits Research, School of Earth Sciences and Engineering and Frontiers Science Center for Critical Earth Material Cycling, Nanjing University, Nanjing, China. ✉e-mail: zyang@ucc.ie

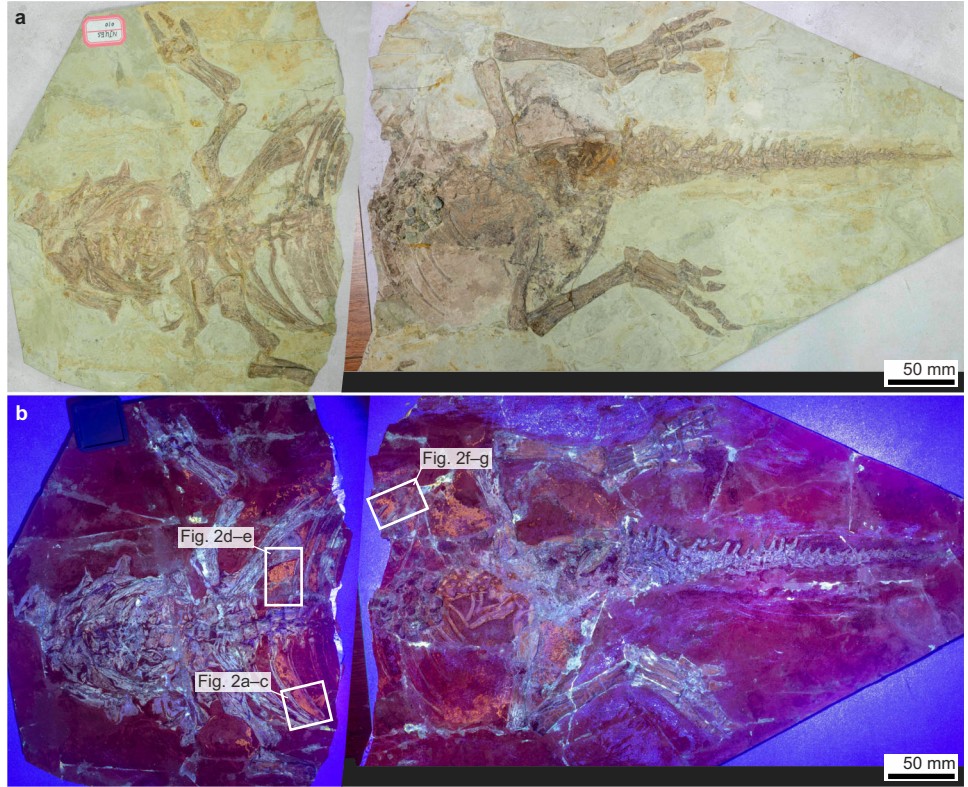

**Fig. 1 | Overview of the *Psittacosaurus* specimen (NJUES-10).** The specimen under natural light (**a**) and UV light (**b**) showing distinct fluorescence hues for bone (cyan) and soft tissues (yellow) against a dark purple sedimentary matrix.

and developmental processes underpin the production of feathers in different archosaurian groups[2,11,12], regardless of whether a single origin applies.

In extant birds, feathers are associated with complex adaptations of the skin, including (1) a thin, pliable epidermis to facilitate motion during flight[13], (2) follicles for the generation and renewal of feathers[14], (3) a shift in the site of epidermal melanogenesis to feather follicles for plumage colouration[15], (4) a dermal muscular system for support and control of feathers[16,17] and (5) a lipid-rich corneous layer for regulation of water- and heat loss[18]. Avian skin is therefore distinct in anatomy and function to scaled reptilian skin, which presumably represents the ancestral condition[14]. Little is known, however, about this evolutionary transition, especially the timing and pattern of acquisition of feather-associated skin modifications[19]. To date, only two studies have examined the skin ultrastructure of basal birds and their close maniraptoran relatives[19,20]. The feathered skin of these taxa had already acquired certain modern characters, including layers of keratin-rich corneocytes that were shed continuously[19] and a dermal system of muscles and connective tissues associated with flight feathers[20]. Resolution of the early evolution of feather-associated skin traits therefore requires analysis of preserved skin in dinosaurs from earlier-diverging clades. Specimens of these taxa are known to preserve scales and non-scaled skin[9,21], but there are few microscopic studies of the preserved remains (but see refs. 22,23).

Here we report ultrastructural preservation of scaled skin from non-feathered body regions of a specimen of *Psittacosaurus* from the Early Cretaceous Jehol Biota of China. The skin is replicated in three dimensions in silica and preserves evidence of epidermal layers, corneocytes and melanosomes. These ultrastructural details indicate that the non-feathered skin of *Psittacosaurus* exhibits the plesiomorphic reptilian condition, demonstrating that early evolution of avian skin traits was restricted to feathered body regions.

## Results

NJUES-10 (School of Earth Sciences and Engineering, Nanjing University, China) is a juvenile specimen of *Psittacosaurus* (see Supplementary Note 1 and 2 for taxonomic assignment and ontogenetic status, respectively, of the specimen), a ceratopsian dinosaur known to possess bristle-like monofilaments on the dorsal tail and scales in non-feathered body regions[21,24,25]. The specimen belongs to the Early Cretaceous (Valanginian–Aptian, ca. 135–120 Ma[26]) Jehol Biota of northeastern China. It preserves a 664 mm long, near-complete, well-articulated skeleton with its ventral side facing upward (Fig. 1 and Supplementary Fig. 1); a well-defined cluster of gastroliths is preserved in the abdomen (Supplementary Fig. 2a, b). In natural light, no preserved soft tissues are evident. Under ultraviolet (UV) light, however, patches of mineralised soft tissues are evident in the torso (shoulder, chest and abdominal flank regions) and along the limbs (the humerus, radius/ulna and femur) (Figs. 1–2 and Supplementary Fig. 2). The soft tissues fluoresce with an orange-yellow hue that is distinct from the cyan hue of the bones, the green hue of the glue (used to adhere various fragments of the slab together) and the dark purple sedimentary matrix. The soft tissues exhibit a distinct texture defined by closely spaced polygons, although locally the tissues are poorly preserved and the polygons, less obvious (Fig. 2c, e, g). The soft tissues are interpreted as the preserved remains of scaled skin based on their close morphological resemblance to extant and fossil archosaurian scales[21,27] and their extensive distribution on the body (Fig. 1 and Supplementary Fig. 2). The preserved scales are tuberculate (i.e., non-overlapping and non-polarised) and polygonal-to-rounded scales. They are mostly ca. 0.8–1.2 mm wide, interspersed with rare larger scales of ca. 1.5–2 mm wide (Supplementary Fig. 3; see also Supplementary Note 3 for additional description of the scale morphology). The former corresponds to basement scales, which in ceratopsian and most other dinosaurs are tuberculate scales (typically 1–10 mm wide) that cover most of the body surface[21,28]. The rare large scales likely

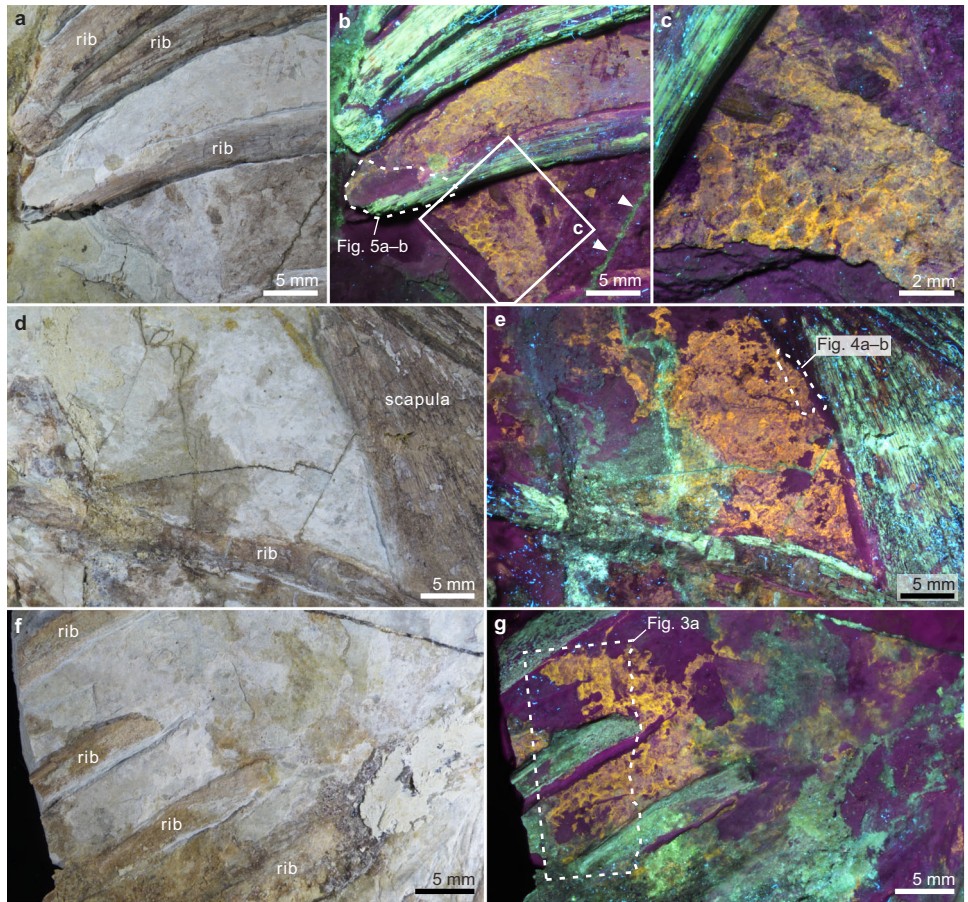

**Fig. 2 | Preserved skin of *Psittacosaurus* (NJUES-10).** Higher magnification views of the soft tissues in the regions indicated in Fig. 1b under natural light (**a**, **d**, **f**) and UV light (**b**, **c**, **e**, **g**); arrowheads in (**b**) indicate glue (green) along a fissure of the slab. Regions defined by dashed lines in (**b**, **e**, **g**) correspond to the samples shown in Figs. 5a–b, 4a–b, 3a, respectively. See also Supplementary Fig. 2 for soft tissues preserved in other body regions.

correspond to feature scales that are individually surrounded by basement scales[21,28]. The preserved *Psittacosaurus* skin represents ventral skin as it drapes over the skeleton (Fig. 3d and Supplementary Fig. 2c, e).

## Ultrastructure of the fossil skin

Samples of the fossil skin were taken from the thorax and abdominal flank regions (Fig. 2b, e, g). SEM analysis of the fossil surface reveals that the skin remains have an amorphous to fine-grained, slightly uneven texture and comprise multiple layers (Supplementary Fig. 4). Hemispherical depressions ca. 5–25 μm in diameter occur locally and comprise aggregates of cubic to subspherical voids, each ca. 0.5–2 μm wide (Supplementary Figs. 4 and 5). These structures are interpreted as external moulds of pyrite framboids that dissolved during diagenesis[29].

Vertical sections of the fossil skin show two major layers (Figs. 3–5). The upper layer is ca. 25–60 μm thick and comprises ca. 10–20 sublayers, each ca. 1.8–2.5 μm thick. Each sublayer is amorphous to nanocrystalline with rare microcrystals up to 7 μm wide (Figs. 3–5 and Supplementary Figs. 6–7). Individual sublayers show undulating upper and lower surfaces in vertical section and typically taper laterally (Figs. 3g–i, 4e, h and 5d–f). Vertical fractures divide the sublayers into fragments, many of which persist laterally for at least 20 μm; rare examples persist for at least 50 μm (Figs. 3e–g, 4d, 5d and Supplementary Fig. 4d). The fragments are often lined by isopachous cement comprising fibrous nanocrystals that are orientated orthogonal to the sublayer surfaces (Fig. 4g and Supplementary Fig. 6f). The total thickness of the upper skin layer and the number of constituent sublayers vary among and within samples. This may reflect the loss of some layers during specimen preparation. There is, however, evidence that some of this variation is original: in certain regions from the thorax where the skin is covered by sediment, only ca. 10 sublayers (combined thickness ca. 25 μm) are present (Fig. 5), while more sublayers (combined thickness ca. 60 μm) are clearly present in the abdominal flank regions (Fig. 3).

This upper skin layer strongly resembles the outermost layer of the epidermis in extant reptiles, i.e., the stratum corneum. In extant reptiles, this skin layer varies in thickness (ca. 10–150 μm) among body regions[30,31]. It comprises a stack of corneocytes—flattened cells typically 15 μm wide that appear elongate and spindle-like in vertical section[32,33]. These cells fuse laterally during development, forming individual syncytial corneocyte layers with partial (in extant crocodiles, chelonians and tuataras) or complete (in extant squamates) loss of cell boundaries[31,32,34,35]. The characteristic layering of the reptile stratum corneum is evident in the upper layer of the fossil skin. Many of the sublayer fragments of the fossil skin (Figs. 3e–g, 4d, 5d and Supplementary Fig. 4d) are wider than individual corneocytes in extant reptiles but are consistent with lateral fusing of corneocyte boundaries. Further, corneocyte thickness is highly conserved in modern amniotes, typically 1–3 μm, irrespective of the total skin thickness[36]. For instance, corneocytes are 1.3–2 μm thick in the crocodile *Crocodylus porosus*[31] and 1.5–2 μm thick in the scutate scales of chicken feet[17]. These thicknesses are comparable to those of the sublayers in the fossil skin, although the isopachous cement, varying from less than 1 μm to ca. 2 μm thick, adds to the original thickness of the fossil structures. As such, the upper layer of the fossil skin is interpreted as

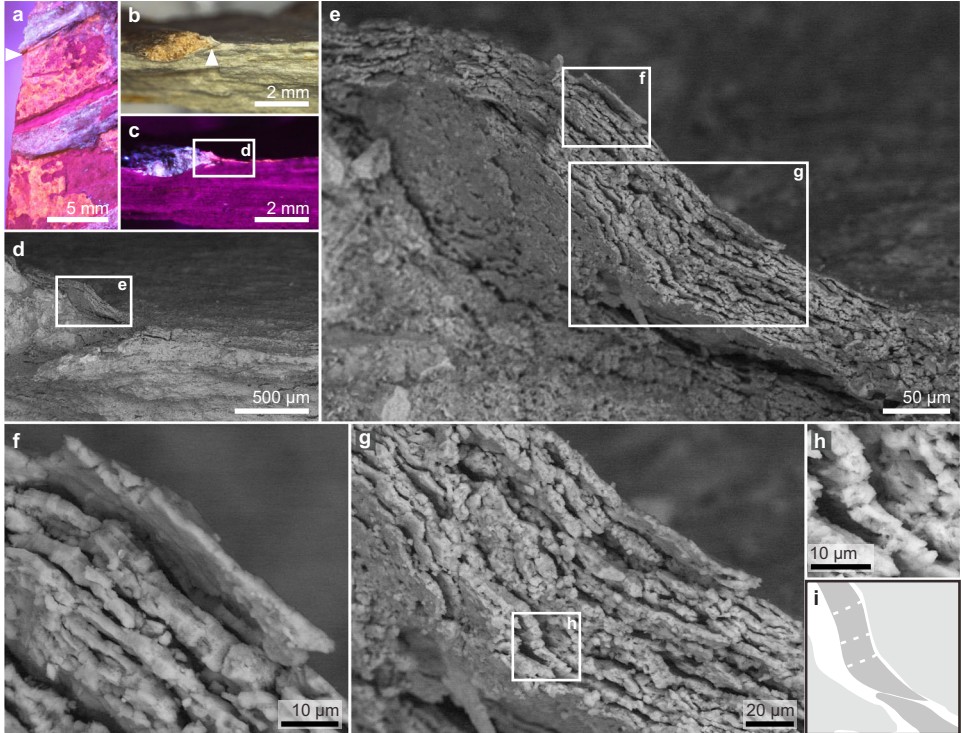

**Fig. 3 | Fossilised *Psittacosaurus* stratum corneum.** Plan view of the fossil surface (**a**, under UV light) and a fractured vertical section (**b** and **c**, under natural light and UV light, respectively) of the fossil skin (sampling location shown in Fig. 2g). Arrowheads in **a** and **b** indicate the same position on the rib bone. **d**–**g** Scanning electron micrographs of the fossil skin showing a layered structure with individual layers that are fragmented laterally. Close-up of the region indicated in **g** (**h**) with interpretive drawing (**i**) highlighting a single sublayer (dark grey in **i**) with tapering lateral tips; light grey shading in **i** denotes over- and underlying sublayers and dashed lines denote fractures.

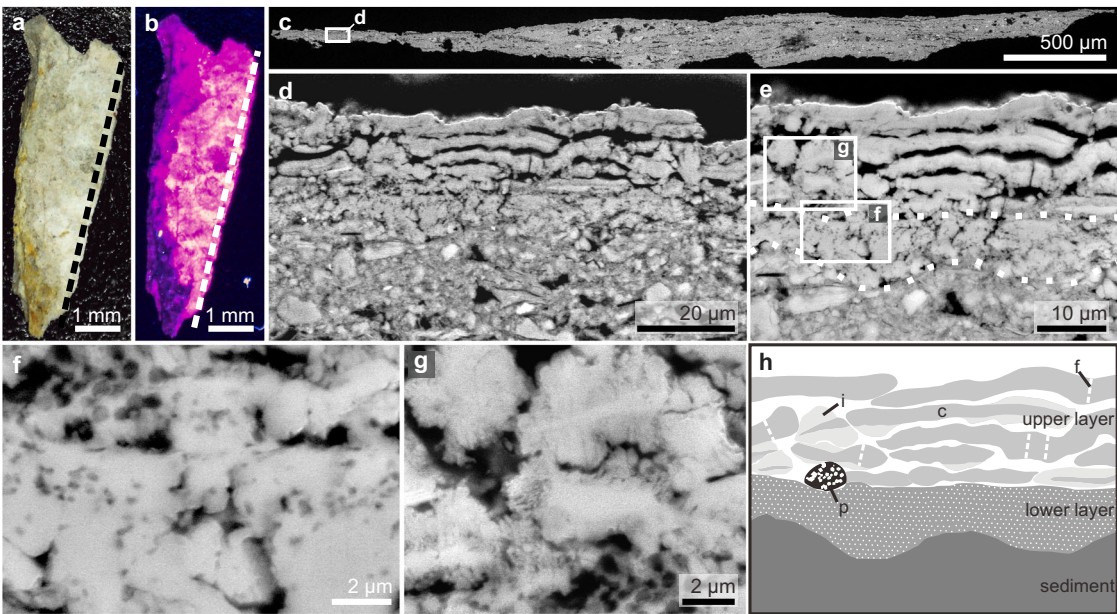

**Fig. 4 | A polished vertical section through the fossil skin showing both upper and lower skin layers.** A fossil skin sample (sampling location shown in Fig. 2e) under natural light (**a**) and UV light (**b**); dashed lines indicate the approximate position of the polished section. **c**–**g** Scanning electron micrographs of the (uncoated) polished section showing the upper and lower fossil skin layers and the underlying sediment. Dashed lines in (**e**) denote the boundaries among the two fossil skin layers and the underlying sediment; note the isopachous cement surrounding sublayer fragments in (**g**). **h** Interpretive drawing of (**e**), showing fossil corneocytes (c) with fractures (f) and isopachous cement (i), and a dissolved pyrite framboid (p); stippled fill for the lower skin layer denotes melanosomes. See also Supplementary Figs. 6–8 for variations in melanosome distribution.

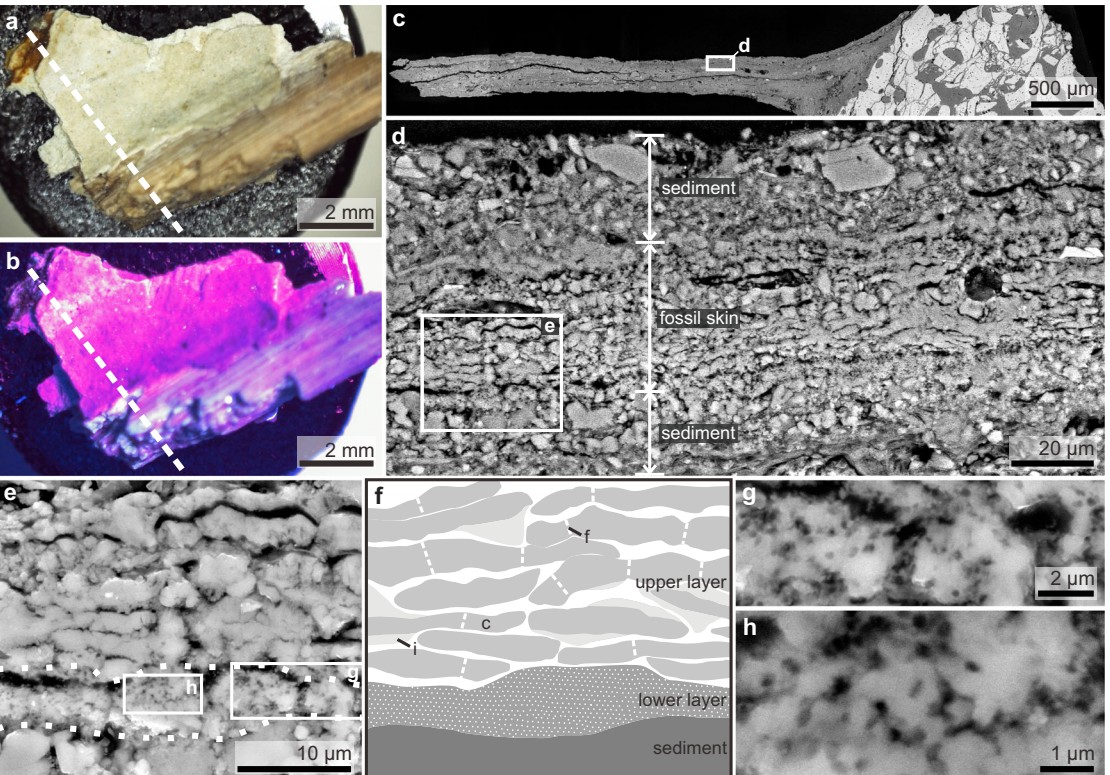

**Fig. 5 | A polished vertical sections through the fossil skin covered by sediment.** A fossil skin sample (sampling location shown in Fig. 2b) under natural light (**a**) and UV light (**b**); dashed lines indicate the approximate position of the polished section in (**c**). Note that in (**b**) the fossil skin (bright yellow) is mostly covered by sediment (purple). **c**–**e** Scanning electron micrographs of an (uncoated) polished section showing the fossil skin sandwiched between layers of sediment; dashed lines in (**e**) denote the boundaries among the upper and lower fossil skin layers and the underlying sediment. **f** Interpretive drawing of (**e**), showing fossil corneocytes (c) with fractures (f) and isopachous cement (i); stippled fill for the lower skin layer denotes melanosomes. **g**, **h** Close-up of the lower skin layer showing mouldic melanosomes. See also Supplementary Figs. 6-8 for variations in melanosome distribution.

the stratum corneum and each sublayer, as a single layer of (laterally fused) corneocytes. The local lateral tapering of the sublayers likely represents remnant cell boundaries.

The lower layer of the fossil skin is 6–20 μm thick and laterally persistent (Figs. 4–5 and Supplementary Figs. 6–8). It is typically amorphous to nanocrystalline with occasional larger microcrystals up to 11 μm wide (Supplementary Fig. 8c). In extant reptiles, the stratum corneum is underlain by the uncornified, typically thinner inner epidermis, and, in turn, the dermis[30,37]. Based on its position and thickness relative to the fossil stratum corneum, the lower layer of the fossil skin is therefore interpreted as the uncornified inner epidermis (with possibly the upper part of the dermis).

Fossil skin samples from the thorax region often show striking moulds of oblate to spheroidal microbodies, each ca. 0.2–0.4 μm wide (Figs. 4–5 and Supplementary Figs. 6–9). The distribution of the microbodies varies: they can occur in the lower skin layer only (Figs. 4–5 and Supplementary Figs. 6 and 8) or in both the upper and lower layers (Supplementary Fig. 7), or they can be absent (Supplementary Fig. 6d). These microbodies fall within the size range of fossil and extant melanosomes[38] and resemble the low aspect ratio melanosomes reported previously in the integumentary structures of pterosaurs[3,38], non-maniraptoran dinosaurs (including *Psittacosaurus*)[25,38] and extant reptiles[39]; in fact, mouldic preservation of melanosomes is common in many fossils[3,25,38]. Critically, the spatial distribution of the microbodies is consistent with that of melanosomes in extant crocodilian scales, where most melanosomes occur in the uncornified epidermis (i.e., in melanocytes) and uppermost dermis (i.e., in melanophores)[39]. In addition, melanosomes derived from melanocytes are incorporated into corneocytes in the stratum corneum, creating macroscopic colour patterns comprising black spots and/or stripes[32,39].

An alternative interpretation is that the microbodies may represent the remains of other skin features, fossil bacteria or abiotic (i.e., taphonomic) artefacts. None of these, however, is likely. Although the microbodies are similar in size to pigment granules in xanthophores in the scales of extant crocodiles[30,39], these granules are dermal, not epidermal[39]. Further, the microbodies do not resemble skin glands in extant reptiles. Reptilian skin glands are multicellular features that are much larger than the fossil microbodies and are restricted to certain body regions[40]. In crocodiles, the closest relative to dinosaurs among extant reptiles, the skin glands are restricted to mandibular, cloacal and dorsal regions[40]. Finally, although bacteria preserved as moulds in various minerals (including silica) have been reported in fossils and taphonomic experiments[41], the microbodies observed in *Psittacosaurus* are unlikely to be fossilised decay bacteria. This is because (1) decay bacteria, where derived externally to the carcass, normally overgrow the tissues[42], whereas the fossil microbodies are clearly internal structures, and (2) even in the case of internally derived decay bacteria, they usually are much more diverse in size (0.1 μm to millimetres) and shape (spirals, baccilliforms and coccoids) than observed for the fossil microbodies[43,44]. Lastly, while microscopic voids can be produced abiotically in crocodile scales during decay[45], such voids exhibit a much wider range of morphologies (a few microns to tens of microns wide and rounded to irregularly shaped) than those of the mouldic microbodies. Collectively, these observations indicate that the microbodies are most parsimoniously interpreted as fossil melanosomes, preserved as three-dimensional moulds.

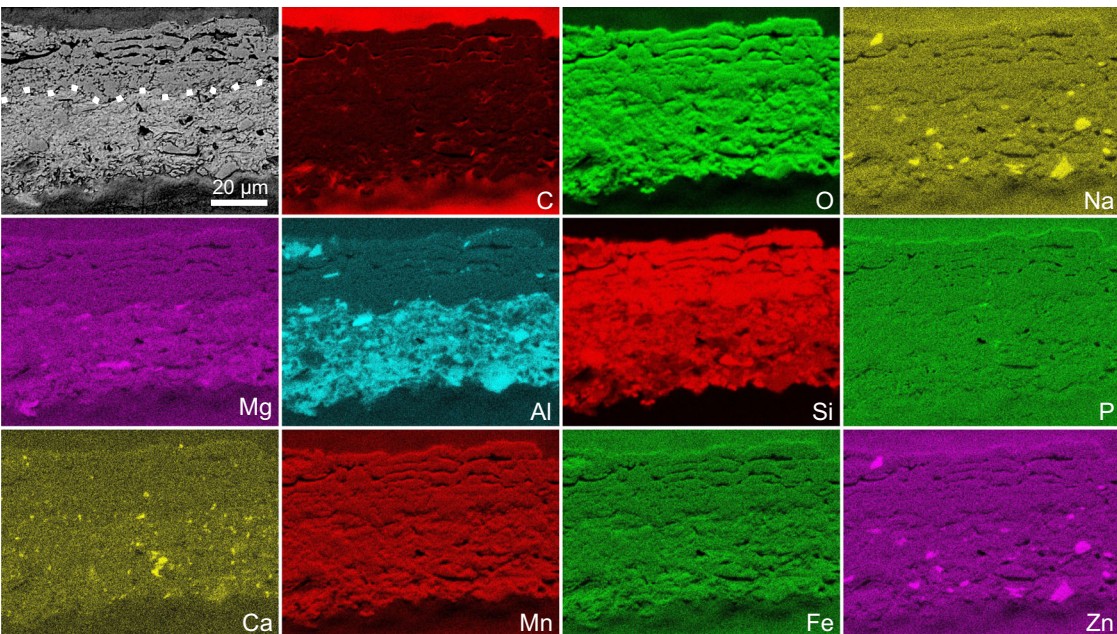

**Fig. 6 | Elemental composition of the fossil skin.** Scanning electron micrograph (first panel) and EDS maps of a resin-embedded vertical section through the fossil skin (approximately the region shown in Fig. 4e). Dashed line in the scanning electron micrograph denotes the boundary between the fossil skin (upper layer) and the underlying sediment.

## Chemistry of the fossil skin

Energy dispersive X-ray spectroscopy (EDS) maps show that the fossil skin is rich in Si and O, but no other elements (Fig. 6), indicating that the tissue is replaced in silica. In contrast, the sedimentary matrix is rich in Si, Al and O, consistent with a composition rich in aluminosilicates; some sedimentary grains are rich in Na, Mg, Ca and Zn. The *Psittacosaurus* bones and clam shrimp shells (from the same bedding plane as the fossil skin) are rich in Ca and P (Supplementary Figs. 10–11), consistent with their in vivo composition, i.e., calcium phosphate[46].

The EDS data are supported by micro-Fourier transform infrared spectroscopy (μ-FTIR), which reveals that IR spectra for the fossil skin are almost identical to that of quartz grains from the sedimentary matrix (Fig. 7 and Supplementary Figs. 12–13). The fossil skin and the quartz grains share three characteristic peaks. The two major peaks at ∽1040 cm$^{-1}$ and ∽770 cm$^{-1}$ correspond to asymmetric and symmetric stretching vibrations of Si–O–Si, respectively[47]. The minor peak at ∽1160 cm$^{-1}$ likely reflects the presence of other cations that may substitute Si within the Si–O–Si lattice[47]. These spectra differ strongly from the spectrum of the embedding resin (Fig. 7). The IR spectra of the fossil skin lack peaks for organic compounds, which is not unexpected for mineralised soft tissues[48].

## Discussion

### Anatomy of non-feathered skin in *Psittacosaurus*

Several specimens of *Psittacosaurus* are known to preserve evidence of the skin[21,24,25,49–52]. Our study reports preserved skin in a new specimen and, critically, provides a comprehensive characterisation of the ultrastructure of the non-feathered scaled skin. Certain preserved ultrastructural features, specifically the thickness of the stratum corneum, the number and syncytial structure of corneocyte layers, and the distribution of melanosomes, provide critical insights into the anatomy of *Psittacosaurus* skin.

The preserved stratum corneum is from the ventral torso and is 25–60 μm thick. This is thicker than that of the non-scaled skin in extant birds (typically 5–10 μm)[33]. The fossil skin thus more closely resembles the stratum corneum of avian scales (20–140 μm)[17,53] and

that of the ventral scales in extant crocodiles (20–160 μm)[39]. The preserved thickness of the fossil stratum corneum is most likely greater than the original thickness in vivo, for two reasons: (1) the corneocytes are enveloped in an isopachous cement; (2) individual corneocyte layers are often separated slightly, indicating desiccation prior to mineralisation (Figs. 3–5 and Supplementary Fig. 4). These features contrast with corneocytes in the scales of extant crocodiles and birds, which show highly ordered, compact stacking of corneocytes with little intercellular space[17,39]. Even so, it is unlikely that taphonomic processes have increased the thickness of the fossil stratum corneum by a factor of five or more. We therefore consider that the original skin thickness fell within the range of thicknesses exhibited by the scaled skin of extant crocodilians and birds, but not the feathered skin of birds. In addition, the preserved skin thickness is consistent with previous observations on hadrosaurs, e.g. the forelimb of *Edmontosaurus*[23], where the stratum corneum is 35–75 μm thick and a saurolophine hadrosaurid[22], where the epidermis is 0.1–0.2 mm thick (thickness of stratum corneum unclear) on the dorsal side of a rib. These hadrosaur skin fossils are preserved organically[22,23] and thus likely subject to different taphonomic processes (and different modifications on skin layer thickness) relative to the silicified skin of NJUES-10.

The number of corneocyte layers in *Psittacosaurus* (ca. 10–20 layers) is lower than that in the scales of extant analogues. In extant crocodiles, the ventral scales of hatchlings and early juveniles of up to a few months old show ca. 10–30 corneocyte layers, similar to *Psittacosaurus*. In these animals, additional corneocytes, however, are added during growth, yielding ca. 40–70 corneocyte layers in juveniles 0.8 m long (i.e., of comparable body length to NJUES-10) and ca. 70–100 corneocyte layers in adults > 2 metres long[31,39]. Avian scales also typically show more corneocyte layers than in *Psittacosaurus*; for instance, there are ca. 50–70 corneocyte layers in the scutate scales and even more in the reticulate scales of chickens[17,54,55]. The number of corneocyte layers in the fossilised skin of hadrosaurs unfortunately remains unclear[22,23].

Compared to the non-scaled skin of extant birds, the skin of *Psittacosaurus* resembles the ventral apterium of zebra finch and blue

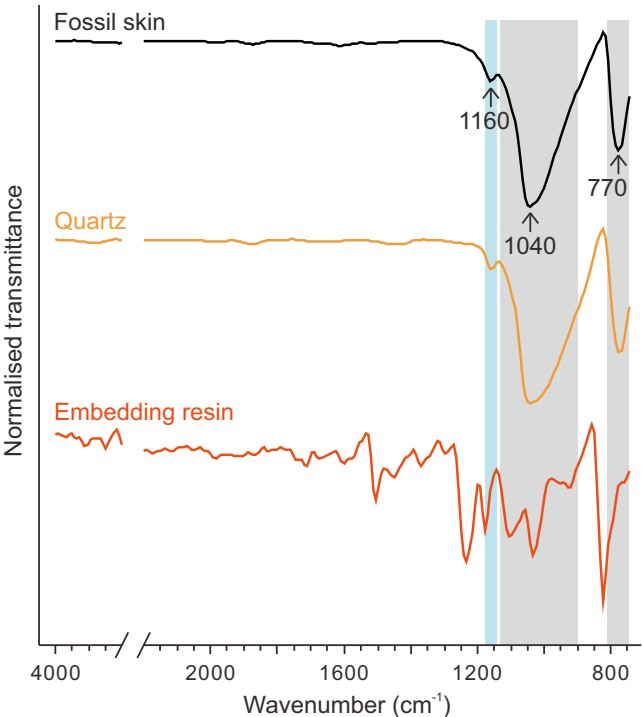

**Fig. 7 | IR signature of the fossil skin.** Representative μ-FTIR transmittance spectra of the fossil skin, quartz grains from the sedimentary matrix and the embedding resin; spot locations at which spectra were collected are shown in Supplementary Fig. 12g. The shaded grey regions indicate Si–O–Si bands and the blue region indicates the band for substitute cations for Si in the Si–O–Si structure[47]. See also Supplementary Figs. 12–13 for μ-FTIR maps of different sample regions and comparison of spectra for the fossil skin derived from each region.

rock pigeon; this is an unexposed, non-feathered, skin region where the stratum corneum is ca. 15–25 corneocytes thick[18,56]. In exposed, non-scaled skin regions of extant birds, however, the stratum corneum is usually thicker; for instance, the naked neck skin in ostriches is ca. 40–45 corneocytes thick[18].

The relatively thin stratum corneum of the *Psittacosaurus* ventral torso, which lacks feathers, therefore informs on the original composition of the skin in vivo. In extant birds, where the stratum corneum comprises keratins (formerly alpha-keratins), it can provide mechanical protection in exposed skin regions (e.g., ostrich neck skin and avian reticulate scales)[17,18]. This function, however, normally requires considerably more corneocyte layers than observed in *Psittacosaurus*[17,18,33]. It is reasonable to assume that the stratum corneum in the exposed ventral skin of *Psittacosaurus* also functioned in mechanical protection. Indeed, the preserved stratum corneum in the abdominal flank (Fig. 3) is thicker than in the thorax region (Fig. 5); this probably reflects the need for protection in more vulnerable body regions, as in extant crocodiles where the scales from the flank and dorsal regions of the torso are thicker than those from the abdomen[37,39]. The low number of fossil corneocyte layers in *Psittacosaurus* is therefore not consistent with a dominant composition of keratins; instead, a function in mechanical protection is feasible for the thin *Psittacosaurus* stratum corneum only if it was originally rich in corneous beta proteins (CBPs). This interpretation is supported by the presence of laterally fused corneocyte boundaries in the *Psittacosaurus* stratum corneum, similar to the syncytial structure in extant reptiles[31,32,34,35]; in contrast, distinct corneocyte boundaries are retained in the keratin-rich avian epidermis[19,57]. The CBP-rich scutate scales of birds also appear to retain the corneocyte boundaries during development[58], which may reflect their different evo-devo pathways compared to reptilian scales[59]. The observed absence of cell

boundaries in individual corneocyte layers of the fossil skin is unlikely to represent a taphonomic artefact, as the cell boundaries between successive corneocyte layers are preserved.

The lateral variation in the distribution of fossil melanosomes within the skin (Figs. 4–5 and Supplementary Figs. 6–9) supports the previous interpretation that the chest of *Psittacosaurus* exhibited macroscopic melanin-based colour patterning[25] (see also Supplementary Note 4 for the skin colour of *Psittacosaurus*). This patterning presumably functioned in display and may be linked to bipedalism: the chest region is unlikely to be visible in a quadrupedal stance[25]. *Psittacosaurus* underwent an ontogenetic shift from quadrupedality to bipedality during the first or second year of life[60]. The specimen studied here may have been a three-year-old individual (Supplementary Note 2); its forelimb-to-hindlimb length ratio (Supplementary Table 1) is consistent with that of (bipedal) (sub-)adults but not (quadrupedal) hatchlings[60]. It is possible that the relatively thin stratum corneum may reflect a reduced demand for mechanical protection in an elevated stance.

**Taphonomy of the fossilised skin**

The siliceous composition of the *Psittacosaurus* skin is considered to reflect primary replication in silica rather than diagenetic overprinting of a precursor mineral phase such as calcium phosphate. This is supported by several factors: (1) the bones of *Psittacosaurus* and the shells of clam shrimp retain their original composition (Supplementary Figs. 10–11), indicating that diagenetic conditions did not dissolve calcium phosphate; (2) the fossil skin has a distinct elemental composition to the sedimentary matrix (Fig. 6), which is inconsistent with wholesale diagenetic overprinting; (3) the fossil skin shows no evidence for calcium phosphate or any other mineral phase commonly associated with mineralised soft tissues[48]; and (4) secondary mineral growth often obscures anatomical features of preserved soft tissues[61] (also see Supplementary Fig. 8 as an example), whereas the *Psittacosaurus* skin is preserved with nanoscale fidelity. The relatively large silica crystals (up to 11 μm wide) incorporated within the skin ultrastructure likely reflect local recrystallisation of primary amorphous silica during diagenesis[62,63]. Critically, these large silica crystals are highly localized (Figs. 4–5 and Supplementary Figs. 6–8), indicating that such recrystallization was not pervasive and did not bias the preservation of melanosome distribution.

Silicified fossils are usually associated with environments characterised by elevated levels of dissolved silica[62,64]. Indeed, the isopachous silica cement surrounding the *Psittacosaurus* corneocytes (Fig. 4g and Supplementary Fig. 6f) indicates the presence of silica-rich pore fluids[65], at least temporarily during diagenesis. These fluids are likely to derive from a sedimentary source. The sediments that host the terrestrial vertebrate fossils of the Jehol Biota are dominated by fine-grained vitric shards and pumice fragments, which are typical products of phreatomagmatic eruptions[66]. These ash particles comprise amorphous silica, which is the most soluble form of silica[67]. Silica solubility would have been further enhanced by the high surface area of the ash particles[68]. The glass-rich sediments of the Jehol Biota may therefore have contributed sufficient silicate ions to promote precipitation, at least locally and on a short-lived basis, during diagenesis. Silica solubility also increases with temperature[69] and, at pH > 9, increases with pH[70]. These factors, however, probably had a limited impact on silicification of the *Psittacosaurus* skin. This is because (1) temperature changes are negligible during very early diagenesis (i.e. prior to substantial burial)[71]; and (2) local pH conditions were likely acidic (due to liberation of abundant organic acids)[72,73] during decay of the carcass and weathering of volcanic ash[74,75], where silica solubility would have changed little with pH[70].

The *Psittacosaurus* specimen preserves a well-articulated skeleton and a well-defined cluster of gastroliths (Supplementary Fig. 2b). These indicate that the carcass had not undergone a bloat-and-float stage,

i.e., where the build-up of decay gases led to flotation of the carcass at the lake surface or in the water column. Such flotation would, in turn, typically lead to disarticulation of the skeleton, loss of some skeletal elements and disintegration of the gastrolith cluster[76,77]. Instead, the carcass likely reached the sediment-water interface soon after death and, critically, remained on the lake floor without extended refloating. The highly articulated, complete nature of the carcass could reflect rapid burial[76], but this scenario is inconsistent with the fine-grained and finely laminated matrix sediment, characteristic of low-energy, suspension-dominated deposits[78]. Instead, refloating may have been prevented by low temperatures and/or deep waters. At low temperatures, low rates of decay and thus of gas production (plus ongoing escape of gases via various orifices) can inhibit the accumulation of sufficient gases for carcasses to become buoyant[79,80]; experimental work has demonstrated that, below about 16 °C, fish carcasses can remain on the water bottom for weeks to months without floating[80]. In deep waters, high hydrostatic pressure compresses decay gases and results in their dissolution, thereby suppressing flotation of the carcasses[79,80]. Consistent with this cold, deep water scenario, previous palaeoclimatic reconstruction for the Jehol Biota indicates cold local conditions with a mean air temperature of $10 \pm 4$ °C[81]; further, although the exact water depths are unknown, the Jehol lakes had steep margins and sufficiently high depth-to-fetch ratios to develop stratification[46,82]. Limited oxygen in the hypolimnion[46] would have prevented scavenging and bioturbation of the carcass.

Although the skin of various animals can survive prolonged periods of decay (months to over a year) postmortem[76,83], silicification of the *Psittacosaurus* skin probably commenced quickly. Recent experimental work has demonstrated that rapid and early silica precipitation can lead to soft tissue preservation within a short time frame postmortem[62]. Hydroxyl, amino and carboxyl groups common in decaying protein-rich tissues[32] can actively bind silicate ion species via hydrogen bonding, electrostatic interactions and cation bridging[62,84]. This process can proceed even at silica concentrations well below saturation, inducing rapid silica precipitation within 24 hours postmortem[62,84]. Given the likely high abundance of these functional groups in the tissue and the availability of dissolved silica, it is likely that silicification of the *Psittacosaurus* skin commenced quickly after arriving at the burial site. Selective preservation of the epidermis of *Psittacosaurus* may reflect (1) faster decay of dermis and muscles; (2) lower concentrations of silica-binding functional groups[62,84] in dermis and muscles, and/or (3) retarded diffusion of silicate ions beyond the silicified epidermis into progressively internal tissue regions (see also Supplementary Note 5 for the lack of feather preservation in NJUES-10).

Silicified animal soft tissues are exceedingly rare in the fossil record[85]. Indeed, replication of ultrastructural details of fossil vertebrate soft tissues in three dimensions in silica has not been reported previously[64,85]. High-fidelity preservation of animal soft tissues is most often associated with replication in calcium phosphate[86]. This mode of preservation is also known to apply to fossils from the Jehol Biota: fossil corneocytes, replicated with nanometre-scale fidelity in calcium phosphate, have been reported in specimens of maniraptoran dinosaurs and a basal bird[19]. Corneocytes of extant birds are rich in keratin, which can become extensively phosphorylated in vivo, thereby serving as a source of phosphate ions during decay[19]. In contrast, the CBP-rich corneocytes of extant reptiles have a very low phosphorus content[36]. Given that the ultrastructural characteristics of the *Psittacosaurus* epidermis are most consistent with those of extant reptiles, the fossil skin may also have had a low phosphorus content in vivo. Finally, phosphatisation usually applies to decay-prone tissues[86,87], because the process requires steep geochemical gradients to be established in the local microenvironment by intense decay[73]. The decay rate of more recalcitrant tissues, such as the *Psittacosaurus* epidermis, may have been too low to establish the necessary geochemical gradients for phosphatisation.

## Evolutionary implications

The scaled skin of reptiles and the feathered skin of birds presumably represent the pleiomorphic and derived conditions, respectively, of the evolutionary transition from scaled to feathered skin[14]. Avian scales, on the other hand, are considered to be secondarily derived structures that evolved after the scale-feather transition, based on both palaeontological and developmental evidence[59,88,89].

Relative to the scaled skin of reptiles and the feathered skin of birds, *Psittacosaurus* skin clearly exhibits the reptilian condition in non-feathered body regions. The fossil stratum corneum is relatively thin and corneocytes show fused cell boundaries, features consistent with a composition rich in CBPs, as in the stratum corneum of extant reptiles[31,32,34,35]. Secondly, the *Psittacosaurus* scales exhibit evidence for melanin-based colour patterning consistent with that in the scales of extant crocodilians. In contrast, the feather-covered epidermis of extant birds is normally unpigmented, with few or no melanosomes[17,90].

Collectively, these findings suggest that *Psittacosaurus* retained the plesiomorphic condition of its scaled reptilian ancestors in non-feathered skin regions. It is reasonable to presume that the skin of feathered body regions, i.e. the tail, exhibited some or all of the modifications related to feather support and movement that characterise the skin of extant birds. This presumed variation in skin structure in *Psittacosaurus* is consistent with spatial partitioning of gene expression, a phenomenon evident during feather development in extant birds due to activation of regional patterning genes[91–93]. For instance, the switch between scale and feather development in the feet of pigeons is governed by the patterning genes Tbx5 and Pitx1, which specify the loci for fore- and hindlimb development, respectively, through region-specific expression[92]. Normally, only Pitx1 is expressed in developing hindlimbs, yielding scaled feet; ectopic expression of Tbx5 and decreased expression of Pitx1 in the hindlimbs cause foot feathering[92]. *Psittacosaurus* may have had similar patterning genes that specified and designated the tail and torso regions for development of different skin structures.

Only two other non-avian dinosaurs exhibit both body scales (sensu ref. 28) and feathers, i.e., the basal neornithischian *Kulindadromeus* and the basal coelurosaurian *Juravenator* (see ref. 94 for a more basal position close to the base of Tetanurae). As in *Psittacosaurus*, these two dinosaurs also had localised distribution of filamentous feather structures and non-feathered body regions bear scales[95,96]. Retention of plesiomorphic skin characters in the non-feathered skin of *Kulindadromeus* and *Juravenator* suggests that the common ancestor of theropods and ornithischians also possessed two coexisting skin conditions: derived, avian-type skin locally in feather tracts and plesiomorphic, reptile-type skin in non-feathered regions.

This hypothesis does not conflict with the possible presence of feathers in the avemetatarsalian ancestor of dinosaurs and pterosaurs[3,4] (but see refs. 8,9), because the latter scenario does not exclude the possibility of coexisting feathers and scales. Indeed, given the reptilian ancestry and the wide taxonomic distribution of scaled skin among ornithischians, sauropods and non-avialan theropods, body scales were almost certainly present in early-diverging dinosaurs and their avemetatarsalian ancestors[9,28].

Limiting skin modifications to feathered body regions was likely a critical factor in the evolution, and increased utilisation, of feathers in dinosaurs and pterosaurs. During the early stages of their evolution, feathers were probably sparse and highly localised on the body, a condition observed in dinosaurs but not in pterosaurs[9,97]. This difference may be an artefact of different sample sizes (feathers have been reported in only three pterosaur specimens[3,4]). Regardless of whether all feathers have a common origin, these early feathers were likely

accompanied by at least some of the skin modifications present in extant birds, given the likely shared genetic and developmental mechanisms for the production of feathers[2,11,12]. Retention of the plesiomorphic skin condition in non-feathered regions would have maintained the essential protective function of the skin against abrasion, desiccation and pathogens[14,27,98]. This may have been vital for the survival of early feathered animals and, critically, the retention of feather genes during early feather evolution. Substantial expansion of feathered regions and loss of body scales may have occurred near the origin of maniraptoriforms, in which the body (excepting the manus, the feet and some part of the legs) was typically covered with feathers[28,96,97] and the underlying epidermis had acquired many modern attributes[19].

## Methods

### Sampling

The studied *Psittacosaurus* specimen NJUES-10 belongs to the fossil collections of the School of Earth Sciences and Engineering, Nanjing University, Nanjing, China. Approval for study of NJUES-10, including destructive sampling and export of samples, was received from the school. The specimen was donated to Nanjing University from a private collection in 2021; data on locality and stratigraphy are not available. The specimen is preserved on a single, fragmented slab that has been glued together along the fissures. The specimen was examined for evidence of preserved skin using a Nikon SMZ25 stereomicroscope coupled with a UV light source (wavelength 365 nm). Under UV light, discontinuous patches of soft tissue can be readily distinguished from the bones, sediment and glue via distinct differences in fluorescence colour. Regions of soft tissue lacking obvious contamination by glue were selected for further analysis. Small samples (most 2–5 mm wide) of soft tissue were dissected from the thorax and abdominal flank regions (Fig. 2) using a scalpel and mounted on carbon tape on aluminium stubs; selected samples were later embedded in epoxy resin and polished.

### Scanning electron microscopy (SEM)

SEM analyses used a JEOL JSM IT-100 variable pressure (VP)-SEM in the School of Biological, Earth and Environmental Sciences, University College Cork. The SEM was equipped with a backscatter detector and a 30 mm² EDS detector. Samples were examined at accelerating voltages of 10–20 kV and a working distance of 10 mm. Most samples were uncoated for SEM analyses in VP mode; selected samples were sputter-coated with Au for high-resolution imaging in high-vacuum mode.

### Micro-attenuated total refection Fourier-transform infrared (µATR-FTIR) spectroscopy

Infrared reflectance spectra were collected from regions of interest in polished vertical sections of the fossil skin in the School of Biological, Earth and Environmental Sciences, University College Cork. Data collection used a Perkin Elmer Spotlight 400i FTIR microscope coupled to a Frontier spectrometer, a dedicated high-resolution ATR Ge imaging accessory and a computer. Collection was via the software SpectrumIMAGE R1.11.2.0016 and the parameters were set as follows: resolution 16 cm⁻¹, 32 scans per pixel, interferometer speed 1.0 cm/s, scan region 4000 cm⁻¹ to 750 cm⁻¹ and pixel size 1.56 µm. The resulting reflectance maps contain an infrared spectrum for each pixel and can be displayed as transmittance or absorbance. A background spectrum was collected prior to each map to account for signal contribution from the instrument and environment. Raw spectra were processed as follows: atmospheric correction (to compensate for water vapour and $CO_2$ contributions) in SpectrumIMAGE R1.11.2.0016 and baseline correction for extracted spectra in SpectraGryph v1.2.16.1 (using the default coarseness and offset values in the advanced baseline correction function). The SpectrumIMAGE and SpectraGryph software are available from https://www.perkinelmer.com/product/spotlight-400-s-w-kit-sp400-lx108895 and www.effemm2.de/spectragryph, respectively.

### Statistics and reproducibility

Image acquisition of the scanning electron micrographs (Figs. 3d–h, 4c–g, 5c–e, g–p, and the first panel of 6 and Supplementary Figs. 4, 6–8, 9c, f, 10a–b, 11a–b and 12a–b, e, h–i) followed convention in the field. These micrographs were obtained as single, unique, images. Repeated acquisition of images in the same region is not standard procedure as it may lead to beam damage.

### Reporting summary

Further information on research design is available in the Nature Portfolio Reporting Summary linked to this article.

## Data availability

The authors declare that the data supporting the findings of this study are available within the paper and the Supplementary Information. NJUES-10 is reposited at the School of Earth Sciences and Engineering, Nanjing University, Nanjing, China and access to the specimen is available upon reasonable request to the curator Baoyu Jiang. Source data are provided with this paper.

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

## Acknowledgements

We are grateful to Shuren Cai for his generous donation of the *Psittacosaurus* specimen to Nanjing University. We thank Shengyu Wang for assistance with identification of clam shrimp fossils, Peter Chung and Pat Meere for their assistance with FTIR work, Naomi O'Reilly for logistical assistance and Tiffany Slater, Valentina Rossi and Richard Unitt for constructive discussion. This work was financially supported by the Government of Ireland Postdoctoral Fellowship (grant no. GOIPD/2021/900) and the Jurassic Foundation grant to Z.X.Y., the National Natural Science Foundation of China (grant no. 42288201) and Fundamental Research Funds for the Central Universities (award no. 0206-14380137) to B.Y.J. and European Research Council Consolidator Grant (grant no. H2020-ERC-COG-101003293-Palaeochem) to M.M.N.

## Author contributions

Z.X.Y. and M.M.N. designed the research. Z.X.Y., B.Y.J. and J.X.X. performed sampling. Z.X.Y. and M.M.N. performed SEM and FTIR analyses. Z.X.Y. and M.M.N. wrote the manuscript with substantial contributions from B.Y.J. and J.X.X.

## Competing interests

The authors declare no competing interests.
