## [Peer Review File · Nature Communications]

Three-dimensional preservation of skin ultrastructure in a feathered dinosaurReviewers' Comments:

Reviewer #1:

Remarks to the Author:

In their manuscript entitled "Three-dimensional preservation of skin ultrastructure in a feathered dinosaur", Zixiao Yang and colleagues describe the microstructure of fossil skin exceptionally preserved in three dimensions in the torso region of a new specimen of the ceratopsian dinosaur *Psittacosaurus* from the Early Cretaceous Jehol Biota of China. The skin is preserved in silica and includes epidermal layers, corneocytes, and melanosomes, providing unique information on these structures in one of the two feathered ornithischian taxa known to date. The authors reveal that the non-feathered scaly skin of this dinosaur's torso exhibits: i) the plesiomorphic reptilian condition of having an original composition rich in corneous beta proteins rather than (alpha-) keratins (as in birds); ii) evidence for melanin-based colour patterning consistent with that in the scales of extant crocodylians; iii) a thin stratum corneum reflecting probably a reduced demand for mechanical protection in the elevated stance of a bipedal subadult individual.

I enjoyed very much reading this contribution on the skin ultrastructure of *Psittacosaurus*. I am admittedly not an expert in skin microstructure as my expertise lies in the morphology and diversity of the scaly integument in non-avian dinosaurs (including *Psittacosaurus*, which I studied or am studying). My non-expert eyes can hopefully offer some helpful insights to improve this study. The manuscript is clear, well-organized, and well-illustrated and despite my limited knowledge of this topic, I was able to follow and understand the text entirely. I am, however, unable to evaluate the robustness of the evidence supporting the presence of three-dimensional skin and its composition. That being said, Maria E. McNamara is one of the few experts in this field and I, therefore, fully rely on her interpretation of the sample under study. Besides, the authors provide many evidence to support their claims and it is clear to me that they have analyzed a sample of scaly skin, whose polygonal basement scales are clearly revealed in UV light. Although the results of this paper are not groundbreaking (they are totally expected from an evolutionary point of view), this study is important because it is the first to provide information on the skin ultrastructure of an ornithischian and a feathered dinosaur from an early-diverging clades, one that includes both scales and protofeathers. I have, consequently, only minor suggestions and remarks to offer and I would personally accept this contribution with minor revisions.

My main remark lies in the absence of information on the skin colour of the *Psittacosaurus* specimen NJUES-10 despite the presence of fossil melanosomes. The shape of these melanosomes is known to provide direct information on the colour of the skin and I would like the authors to discuss this aspect in their paper. What colour(s) was (were) present in the torso and limbs of NJUES-10 based on the shape and concentration of the melanosomes in the sample analyzed by the authors? Are there any patterns such as stripes? How does it compare to that of the Frankfurt specimen SMF R 4970 (which shows evidence of countershading) and the two specimens PKUPV1050 and PKUP V1051 sampled by Li et al. (2014)?

Is it unfortunate that NJUES-10 does not preserve the filamentous structure present in the tail of SMF R 4970. This is, actually, not the first *Psittacosaurus* specimen preserving skin but no monofilament. Similarly, two exquisitely preserved specimens PKUPV1050 and PKUP V1051 illustrated by Li et al. (2014) also show this condition. The monofilaments are quite thick and appear to be robust in SMF R 4970, which made me wonder whether they may have been only present in some *Psittacosaurus* individuals (the male or the female) or species (less likely). Could the authors discuss the absence of monofilaments in NJUES-10 based on the taphonomical scenario they discuss in their section "Taphonomy of the fossilised skin"? Is it normal that these structures decay faster than the scaly integument? Any information on this topic would be particularly interesting.

The description of the preserved integument in NJUES-10 is also rather limited according to me. Can the authors provide, in the text or the supplementary information, additional information on the skin morphology of this specimen and how it compares to the more extensive skin of SMF R 4970? Can they observe any variation in the morphology and size of the basement scales along the torso and the limbs where the skin is preserved? The shoulder region of the forelimb of SMF R 4970, for instance, includes feature scales whereas a hexagram pattern with triangular scales is present in the posterior portion of the upper limbs. It would, therefore, be interesting to know if NJUES-10 shows a similar pattern or if there were some intraspecific variations in skin morphology in *Psittacosaurus*.

The authors assume that the relatively thin stratum corneum of the *Psittacosaurus* ventral torso “may reflect a reduced demand for mechanical protection in an elevated stance”, therefore arguing that NJUES-10, though to be around three years old, probably represents a bipedal subadult individual (as a quadrupedal adult would have a much thicker stratum corneum). This information is quite interesting and I would invite the authors to provide this information in the abstract if the word limit allows it.

Other minor remarks:

- Could the authors provide, in the result section, information on the length of the specimen and the probable stages of the sediments it was found?
- The scaly skin of NJUES-10 can be described as having tuberculate (i.e., non-overlapping and non-polarised) and polygonal basement scales. The authors can refer to my paper (Hendrickx et al. 2022) for a definition of these scales in dinosaurs.
- I do not think that the feathers of *Kulindadromeus* can be described as simple. They are rather complex, with filaments associated with a basal plate and ribbon-shaped structures (Godefroit et al. 2020).

I provided a few additional remarks and corrections in the Word file, which I invite the authors to check.

Sincerely,

Christophe Hendrickx, San Miguel de Tucumán, the 15th of November 2023

References associated with this review

- Godefroit, P., Sinitsa, S. M., Cincotta, A., McNamara, M. E., Reshetova, S. A. & Dhouailly, D. 2020. Integumentary structures in *Kulindadromeus zabaikalicus*, a basal neornithischian dinosaur from the Jurassic of Siberia. In: Foth, C. & Rauhut, O. W. M. (eds) *The Evolution of Feathers: From Their Origin to the Present*. Springer International Publishing, Cham, Fascinating Life Sciences, 47–65., doi: 10.1007/978-3-030-27223-4_4.
- Hendrickx, C., Bell, P. R., Pittman, M., Milner, A. R. C., Cuesta, E., O'Connor, J., Loewen, M., Currie, P. J., Mateus, O., Kaye, T. G. & Delcourt, R. 2022. Morphology and distribution of scales, dermal ossifications, and other non-feather integumentary structures in non-avian theropod dinosaurs. *Biological Reviews*, 97, 960–1004, doi: 10.1111/brv.12829.
- Li, Q., Clarke, J. A., Gao, K.-Q., Zhou, C.-F., Meng, Q., Li, D., D'Alba, L. & Shawkey, M. D. 2014. Melanosome evolution indicates a key physiological shift within feathered dinosaurs. *Nature*, 507, 350–353, doi: 10.1038/nature12973.

Reviewer #2:

Remarks to the Author:

Yang et al. present morphological, histological and elemental evidence for high fidelity preservation of multiple epidermal layers (stratum corneum, lower epidermis) in the ornithischian dinosaur, Psittacosaurus. Firstly, I congratulate the authors on their discovery (something that I've been hoping to find for many years!) and the coherent, well-written manuscript. While I am generally in favour of the findings and evidence presented, I do think there are several areas that need further exploration, which will strengthen the current manuscript and make it a suitable publication for Nature Communications. My main points for considering are detailed below, but additional edits/comments can be found in the attached marked-up version of the manuscript.

Firstly, the authors regard the filamentous structures on the tail of Psittacosaurus as homologous with theropod feathers. This, however, remains controversial, and homology has not been conclusively demonstrated. Stating in the title that it is a 'feathered dinosaur' certainly seems overreaching. The findings are interesting enough and stand on their own without the need to invoke feathers at this point. While the authors can 'side' with whomever they like, I suggest that they also acknowledge the more circumspect publications on this topic in the remainder of the manuscript (e.g. Barrett et al., 2015; Campione et al. 2020).

Lines 118-123: The authors note variation in the thickness of the upper layer. There is well-documented variation in the thickness of stratum corneum across the body in modern reptiles, which the authors are clearly aware of. It would therefore be helpful if you could explicitly state where 'thin' vs 'thick' stratum corneum appear in the present specimen and whether it conforms to changes in thickness in modern reptiles, or even birds. The variation in birds is naturally correlated to presence/absence of feathers, so this could be elaborated on in the Discussion (see also comment below re lines 251-262).

Lines 172-185: While I have no a priori concerns about the interpretation of microbodies, the figures do not do them justice: I cannot see the 'spheroidal' nature of any of them; rather, they show up as hazy black specks in both the figures and supp info. Since these images are the only 'evidence', some higher res images are strongly recommended.

Line 209: I suspect the FTIR analysis was performed in the hope of identifying organic traces. If so, It might be worth mentioning that no peaks could be attributed to stretch vibrations for organic material, which I actually find a little surprising given the apparent fidelity of the skin layers! Something else to discuss perhaps, from a taphonomic perspective, although I understand space is tight in this format. Micron-scale preservation of features seems to be fairly regularly reported for silicified invertebrate fossils, but not, as far as I'm aware, in vertebrates.

Lines 251-262: the authors compare the thickness of stratum corneum in Psittacosaurus to the 'non-scaled' skin of birds and the ventral scales of crocodilians. This seems like an odd comparison, given that there is a well-established relationship in birds, which have thinner stratum corneum in non-feathered regions, whereas it is thicker on the scaled legs (see various Stettenheim references for e.g.). Similarly, since the current paper relates to scales of Psittacosaurus, it would seem more appropriate to comment on scales in birds rather than the non-scaled regions. Further consideration will then also need to be made considering that avian scales are non-homologous with reptilian scales. Subsequently, it feels like an 'apples to oranges' comparison at present. Secondly, while the comparisons are topographically equivalent between crocs and Psittacosaurus (i.e. ventral scales), it

would also be worth commenting on the variation between body regions in crocs. I am unsure how absolute values for stratum corneum thickness vary with ontogeny in modern reptiles and birds, but I encourage the authors to explore and comment on this potential variation as well as it has strong bearing on the current paper (i.e. a 3-6m long *C. porosus* may not be a suitable comparison with a juvenile *Psittacosaurus*!). Considering both of these aspects would greatly strengthen their argument. The conclusion on line 262 that “the original skin thickness to have fallen within the range of thicknesses exhibited by extant crocodylians and not birds” therefore needs further consideration. An additional sentence comparing your values to those reported in hadrosaurs by Fabbri et al (2019) and Barbi et al (2019) would also be relevant here. The latter paper was not cited, but should be (see also line 49).

Barbi, M., Bell, P.R., Fanti, F., Dynes, J.J., Kolaceke, A., Buttigieg, J., Coulson, I.M. and Currie, P.J., 2019. Integumentary structure and composition in an exceptionally well-preserved hadrosaur (*Dinosauria: Ornithischia*). *PeerJ*, 7, p.e7875.

Lines 341-451: The authors state that the specimen had not undergone “bloat and float” based on the articulation of element and the intact mass of gastroliths. While I agree that the specimen did not remain exposed and likely hadn’t been dead for long, it is difficult to imagine how a terrestrial dinosaur could end up within a fine-grained, presumably deep lacustrine environment had it not floated out to its final resting place. A bloat and float scenario would seem a better explanation for why we have an animal (1) preserved belly-up (2) in a quiet, deep water setting. While I agree the degree of articulation may appear at odds with the findings of, say, Syme and Salisbury (2014), the other evidence is more compelling in this case. Perhaps of interest, the overall proportions of a juvenile *Psittacosaurus* (especially this one) do recall the proportions of ankylosaurs, which are frequently overturned, presumably also as a result of bloat and float (see Mallon et al. 2018 *Pal. Pal. Pal.*). One other question pertinent to this argument is: is the specimen actually preserved belly up, or has it just been prepared that way? Although the provenance of the specimen is unknown, there should (hopefully) be some sedimentary markers that indicate way up. This did not come across clearly in the results, which simply state “It preserves a near-complete, well-articulated skeleton in ventral aspect”. The orientation should be backed-up and confirmed as it is relevant to the taphonomy of the specimen. For the record, I have no issue with the other aspects of their taphonomic interpretation.

Lines 398-399: The authors state that *Psittacosaurus* “clearly exhibits the reptilian condition”, but as mentioned, there needs to be a more detailed comparison to the scales of birds before this statement can be considered.

I hope these comments are helpful.

Regards,
Phil Bell

Response to reviewers

We sincerely thank all the reviewers for their insightful and constructive comments and suggestions on our manuscript. Following the reviewers' comments and suggestions, we have carefully revised the manuscript. We have also addressed all the comments and questions, which are detailed below with our responses in red.

REVIEWER COMMENTS

Reviewer #1 (Remarks to the Author):

Comment 1: In their manuscript entitled "Three-dimensional preservation of skin ultrastructure in a feathered dinosaur", Zixiao Yang and colleagues describe the microstructure of fossil skin exceptionally preserved in three dimensions in the torso region of a new specimen of the ceratopsian dinosaur Psittacosaurus from the Early Cretaceous Jehol Biota of China. The skin is preserved in silica and includes epidermal layers, corneocytes, and melanosomes, providing unique information on these structures in one of the two feathered ornithischian taxa known to date. The authors reveal that the non-feathered scaly skin of this dinosaur's torso exhibits: i) the plesiomorphic reptilian condition of having an original composition rich in corneous beta proteins rather than (alpha-) keratins (as in birds); ii) evidence for melanin-based colour patterning consistent with that in the scales of extant crocodylians; iii) a thin stratum corneum reflecting probably a reduced demand for mechanical protection in the elevated stance of a bipedal subadult individual.

I enjoyed very much reading this contribution on the skin ultrastructure of Psittacosaurus. I am admittedly not an expert in skin microstructure as my expertise lies in the morphology and diversity of the scaly integument in non-avian dinosaurs (including Psittacosaurus, which I studied or am studying). My non-expert eyes can hopefully offer some helpful insights to improve this study. The manuscript is clear, well-organized, and well-illustrated and despite my limited knowledge of this topic, I was able to follow and understand the text entirely. I am, however, unable to evaluate the robustness of the evidence supporting the presence of three-dimensional skin and its composition. That being said, Maria E. McNamara is one of the few experts in this field and I, therefore, fully rely on her interpretation of the sample under study. Besides, the authors provide many evidence to support their claims and it is clear to me that they have analyzed a sample of scaly skin, whose polygonal basement scales are clearly revealed in UV light. Although the results of this paper are not groundbreaking (they are totally expected from an evolutionary point of view), this study is important because it is the first to provide information on the skin ultrastructure of an ornithischian and a feathered dinosaur from an early-diverging clades, one that includes both scales and protofeathers. I have, consequently, only minor suggestions and remarks to offer and I would personally accept this contribution with minor revisions.

Response: Many thanks for your appreciation of our work and your critical comments. We have made the suggested revisions and responded in detail to each point below. Please note that the line numbers in the red text correspond to the clean version of the revised manuscript.

Comment 2: My main remark lies in the absence of information on the skin colour of the *Psittacosaurus* specimen NJUES-10 despite the presence of fossil melanosomes. The shape of these melanosomes is known to provide direct information on the colour of the skin and I would like the authors to discuss this aspect in their paper. What colour(s) was (were) present in the torso and limbs of NJUES-10 based on the shape and concentration of the melanosomes in the sample analyzed by the authors? Are there any patterns such as stripes? How does it compare to that of the Frankfurt specimen SMF R 4970 (which shows evidence of countershading) and the two specimens PKUPV1050 and PKUP V1051 sampled by Li et al. (2014)?

Response: We thank the reviewer for these intriguing questions. A correlation between melanosome geometry and colour has been established only for mammalian hair and maniraptoran feathers (Li et al. 2014). According to the data published in that study, there is no significant correlation between melanosome geometry and coloration in reptilian scales. We can thus comment only in very broad terms regarding the coloration of the *Psittacosaurus* integument. The observed lateral variation in melanosome distribution in the skin suggests at least millimetre-scale spatial variation in skin tone. However, the small size of the samples analyzed renders it impossible to comment on the granularity of that variation in tone, or the extent of the variation in tone across the body. We can, however, say that the variation in melanosome abundance is consistent with colour patterning but does not exclude the possibility that variations in tone were highly localised.

We have added the discussion on the skin colour of NJUES-10 and comparison with other specimens in the section “Skin colour of *Psittacosaurus*” in Supplementary Information. We have also added a new figure (Supplementary Fig. 9) showing the spatial extent of possible skin tone variation.

Reference

Li, Q. et al. Melanosome evolution indicates a key physiological shift within feathered dinosaurs. *Nature* 507, 350–353 (2014).

Comment 3: Is it unfortunate that NJUES-10 does not preserve the filamentous structure present in the tail of SMF R 4970. This is, actually, not the first *Psittacosaurus* specimen preserving skin but no monofilament. Similarly, two exquisitely preserved specimens PKUPV1050 and PKUP V1051 illustrated by Li et al. (2014) also show this condition. The monofilaments are quite thick and appear to be robust in SMF R 4970, which made me wonder whether they may have been only present in some *Psittacosaurus* individuals (the male or the female) or species (less likely). Could the authors discuss the absence of monofilaments in NJUES-10 based on the taphonomical scenario they discuss in their section “Taphonomy of the fossilised skin”? Is it normal that these structures decay faster than the scaly integument? Any information on this topic would be particularly interesting.

Response: Agreed. We thank the reviewer for the suggestion and we have added discussion in the section “The lack of feather preservation in NJUES-10” in the Supplementary Information. We did not include this discussion in the main text because we feel the main focus of the study is on the preserved skin rather than the lack of feathers. Regarding the lack of preserved filaments: we feel that this may reflect the mode of preservation of the soft tissues in this specimen. Almost all fossil feathers are preserved as organic (carbonaceous) remains. There are no known examples of feathers preserved as three-dimensional authigenic replacements in silica (as per the skin). It is possible that feathers have an extremely low likelihood of preserving in this way.

Comment 4: The description of the preserved integument in NJUES-10 is also rather limited according to me. Can the authors provide, in the text or the supplementary information, additional information on the skin morphology of this specimen and how it compares to the more extensive skin of SMF R 4970? Can they observe any variation in the morphology and size of the basement scales along the torso and the limbs where the skin is preserved? The shoulder region of the forelimb of SMF R 4970, for instance, includes feature scales whereas a hexagram pattern with triangular scales is present in the posterior portion of the upper limbs. It would, therefore, be interesting to know if NJUES-10 shows a similar pattern or if there were some intraspecific variations in skin morphology in Psittacosaurus.

Response: Agreed. We have provided additional description of the preserved scales in both the main text (Results section, lines 93–100) and the Supplementary Information (section “Variation in preserved scale morphology in NJUES-10” and Supplementary Fig. 3).

Comment 5: The authors assume that the relatively thin stratum corneum of the Psittacosaurus ventral torso “may reflect a reduced demand for mechanical protection in an elevated stance”, therefore arguing that NJUES-10, though to be around three years old, probably represents a bipedal subadult individual (as a quadrupedal adult would have a much thicker stratum corneum). This information is quite interesting and I would invite the authors to provide this information in the abstract if the word limit allows it.

Response: Agreed. We have added this information in the abstract (lines 18–20).

Other minor remarks:

Comment 6: Could the authors provide, in the result section, information on the length of the specimen and the probable stages of the sediments it was found?

Response: Agreed. We have provided the information in the Results section (lines 77–80).

Comment 7: The scaly skin of NJUES-10 can be described as having tuberculate (i.e., non-overlapping and non-polarised) and polygonal basement scales. The authors can refer to my paper (Hendrickx et al. 2022) for a definition of these scales in dinosaurs.

Response: Agreed. We have revised accordingly in the Results section (lines 93–94) and in the Supplementary information.

Comment 8: I do not think that the feathers of *Kulindadromeus* can be described as simple. They are rather complex, with filaments associated with a basal plate and ribbon-shaped structures (Godefroit et al. 2020).

Response: Agreed. The feathers of *Kulindadromeus* are now described as filamentous instead of simple (line 480).

I provided a few additional remarks and corrections in the Word file, which I invite the authors to check.

Comments in the attached document:

Comment 9: (Line 18) I would suggest to add the fact that this is an avian trait, because most people are unaware of this.

Response: Agreed and we have revised accordingly (line 18).

Comment 10: (Line 63) Please provide the stages if possible.

Response: Agreed. We have provided the stages in the revised manuscript (line 78).

Comment 11: (Lines 64–65) Please provide information on the length of the specimen.

Response: Agreed. We have provided the length in the revised manuscript (line 79–80).

Comment 12: (Line 68) Please be more specific. What limb, what part of the torso?

Response: Agreed. The soft tissues are evident in the trunk (shoulder, chest and abdominal flank regions) and along the limbs (arms and femurs). We have revised accordingly (lines 83–85).

Comment 13: (Line 78) You can also cite my paper, cited in [81].

Response: Agreed and we have revised accordingly (line 99).

Comment 14: (Line 79) Do you note any variation in the size and morphology the basement scales? Have you noticed feature scales, typically present in the upper limb of *Psittacosaurus*?

Response: The preserved basement scales vary in size and morphology across the specimen. The preserved features scales are rare and occur on the flank of the lower abdomen. We have provided additional description of the preserved scales in both the main text (Results section, lines 93–100) and the Supplementary Information (section “Variation in preserved scale

morphology in NJUES-10" and Supplementary Fig. 3).

Comment 15: (Line 80) Does it apply to the limbs as well? Is it the ventral skin, the one from the limbs?

Response: Yes, the preserved skin layer drapes over the limbs as well. We have revised the Results section (line 100–102) and Supplementary Fig. 2c and e to make this clear.

Comment 16: (Line 92) Please specify in here from what part of the body the skin sample was taken from.

Response: Agreed and we have revised accordingly (lines 114–115).

Comment 17: (Lines 176–186) Because you assume the presence of fossil melanosomes in this part of the body, isn't there anything to say about the putative colour of this body part of *Psittacosaurus*? Can you perhaps talk a bit more about their shape and concentration, and what it tell us about the colour of the skin?

Response: This comment is effectively the same as Comment 2. Please see our response to Comment 2.

Comment 18: (Lines 301–304) This is quite interesting and I would suggest to provide this result in the abstract.

Response: Agreed and we have revised accordingly (lines 18–20).

Comment 19: (Lines 376–377) What about the filamentous structures?

Response: This comment is effectively the same as Comment 3. Please see our response to Comment 3.

Comment 20: (Line 424) Unlike *Juravenator*, the feathers of *Kulindadromeus* are far from simple as they include filaments associated with a basal plate and ribbon-shaped structures (Godefroit et al., 2020).

Godefroit, P. et al. Integumentary structures in *Kulindadromeus zabaikalicus*, a basal neornithischian dinosaur from the Jurassic of Siberia. in *The Evolution of Feathers: From Their Origin to the Present* (eds. Foth, C. & Rauhut, O. W. M.) 47–65 (Springer International Publishing, 2020). doi:10.1007/978-3-030-27223-4_4.

Response: Agreed. The feathers of *Kulindadromeus* are now described as filamentous instead of simple (line 480).

Comment 21: (Line 434) This has to be specified given that all avemetatarsalian/dinosaurs have reticulate scales on their hand/feet. See my Biological review paper (Hendrickx et al., 2022).

Response: Agreed and we have revised accordingly (line 477).

Comment 22: (Lines 439–440) This indeed appears to be the case in *Juravenator*, but not in pterosaurs, which are currently the earliest branching archosaurs preserving feathers (if they are indeed “true” feathers homologous to those of theropods).

Response: Agreed. Indeed the condition (of early feathers being sparse and highly localised) has been observed in dinosaurs but not in pterosaurs. This difference may be an artefact of different sample sizes (feathers have been reported in only three pterosaur specimens). Nevertheless we have added this to the Discussion (lines 495–497).

Comment 23: (Lines 460–461) Please specify in here from what precise zone of the bone the samples were taken.

Response: Agreed and we have revised accordingly (lines 521–522).

Comment 24: (Lines 434) I would suggest to provide information on the place where the samples were examined using a SEM.

Response: Agreed and we have revised accordingly (lines 526–527).

Sincerely,

Christophe Hendrickx, San Miguel de Tucumán, the 15th of November 2023

References associated with this review

Godefroit, P., Sinitsa, S. M., Cincotta, A., McNamara, M. E., Reshetova, S. A. & Dhouailly, D. 2020. Integumentary structures in *Kulindadromeus zabaikalicus*, a basal neornithischian dinosaur from the Jurassic of Siberia. In: Foth, C. & Rauhut, O. W. M. (eds) *The Evolution of Feathers: From Their Origin to the Present*. Springer International Publishing, Cham, Fascinating Life Sciences, 47–65., doi: 10.1007/978-3-030-27223-4_4.

Hendrickx, C., Bell, P. R., Pittman, M., Milner, A. R. C., Cuesta, E., O’Connor, J., Loewen, M., Currie, P. J., Mateus, O., Kaye, T. G. & Delcourt, R. 2022. Morphology and distribution of scales, dermal ossifications, and other non-feather integumentary structures in non-avian theropod

dinosaurs. *Biological Reviews*, 97, 960–1004, doi: 10.1111/brv.12829.

Li, Q., Clarke, J. A., Gao, K.-Q., Zhou, C.-F., Meng, Q., Li, D., D’Alba, L. & Shawkey, M. D. 2014. Melanosome evolution indicates a key physiological shift within feathered dinosaurs. *Nature*, 507, 350–353, doi: 10.1038/nature12973.

Reviewer #2 (Remarks to the Author):

Comment 1: Yang et al. present morphological, histological and elemental evidence for high fidelity preservation of multiple epidermal layers (stratum corneum, lower epidermis) in the ornithischian dinosaur, *Psittacosaurus*. Firstly, I congratulate the authors on their discovery (something that I’ve been hoping to find for many years!) and the coherent, well-written manuscript. While I am generally in favour of the findings and evidence presented, I do think there are several areas that need further exploration, which will strengthen the current manuscript and make it a suitable publication for *Nature Communications*. My main points for considering are detailed below, but additional edits/comments can be found in the attached marked-up version of the manuscript.

Response: We thank the reviewer for the generous and helpful feedback. We have made the suggested revisions and responded in detail to each point below. Responses to the additional comments (from the marked-up version of the manuscript attached by the reviewer) are at the end of this document. Please note that the line numbers in the red text correspond to the clean version of the revised manuscript.

Comment 2: Firstly, the authors regard the filamentous structures on the tail of *Psittacosaurus* as homologous with theropod feathers. This, however, remains controversial, and homology has not been conclusively demonstrated. Stating in the title that it is a ‘feathered dinosaur’ certainly seems overreaching. The findings are interesting enough and stand on their own without the need to invoke feathers at this point. While the authors can ‘side’ with whomever they like, I suggest that they also acknowledge the more circumspect publications on this topic in the remainder of the manuscript (e.g. Barrett et al., 2015; Campione et al. 2020).

Response: The term “feather” has been equivocal in the literature. It has been used to refer to a variety of integumentary structures, including (1) the feathers of birds, (2) the filamentous and pennaceous integumentary structures of theropod dinosaurs, (3) the pennaceous integumentary structures of avemetatarsalians and (4) the filamentous and pennaceous integumentary structures of avemetatarsalians.

To avoid confusion, we have clarified our definition of “feather” in the Introduction section (lines 30–38) in the revised manuscript. We have followed the approach that defines “feather” in its broadest sense, i.e., the filamentous and pennaceous integumentary structures of avemetatarsalians. We use “group” + “feather” to refer to feathers of specific groups (e.g.

“avian feathers”, “theropod feathers”, “pterosaur feathers”, etc.). We feel this approach is appropriate as it does not necessarily imply a single origin. Similarly, the term “scale” has been widely used to refer to both reptilian scales and avian scales that evolved independently.

We agree that it is possible that feathers evolved independently in theropods, ornithischians and pterosaurs. However, given the shared morphology and histology of these feather structures, and the likely shared genomic heritage and shared pattern of developmental stages of these organisms, we believe that a common origin is more likely. At the very least, the striking morphological and histological similarities among the early feathers of the different groups, and the deep homology of extant reptilian scales and avian feathers, strongly suggest a very similar, if not identical, developmental process underlying the production of these early feathers, regardless of whether they arose from a single origination event or not.

We therefore feel that the scales and feathers of *Psittacosaurus* are extremely relevant to understanding the early evolution of feathers as a whole, particularly the scale-feather transition, despite the possibility of convergent feather evolution. Critically, our finding of reptile-style skin histology in non-feathered regions of *Psittacosaurus*, and the previous finding of tail feathers, are two key pieces of evidence that collectively suggest that the acquisition of skin modifications during the scale-feather transition may initially have been localised to feathered regions.

As such, we think it is critical to refer to feathers early in the paper, and that it is appropriate to refer to *Psittacosaurus* as a feathered dinosaur. We acknowledge, however, controversy regarding the homology of the *Psittacosaurus* feathers with theropod feathers and have revised the Introduction (lines 39–47) and Discussion (the section “Evolutionary implications”, lines 497–500) accordingly.

Comment 3: Lines 118-123: The authors note variation in the thickness of the upper layer. There is well-documented variation in the thickness of stratum corneum across the body in modern reptiles, which the authors are clearly aware of. It would therefore be helpful if you could explicitly state where ‘thin’ vs ‘thick’ stratum corneum appear in the present specimen and whether it conforms to changes in thickness in modern reptiles, or even birds. The variation in birds is naturally correlated to presence/absence of feathers, so this could be elaborated on in the Discussion (see also comment below re lines 251-262).

Response: Agreed. We have clarified the body regions where the thickness of stratum corneum varies (lines 144–148). We have also added comparison of this thickness variation across the body with extant crocodiles (lines 325–329 in the Discussion section “Anatomy of non-feathered skin in *Psittacosaurus*”) but not birds, because the former is a better extant analogue in terms of anatomy (having scales in body regions equivalent to those of *Psittacosaurus*) and ecology (having a terrestrial rather than volant lifestyle).

Comment 4: Lines 251-262: the authors compare the thickness of stratum corneum in *Psittacosaurus* to the ‘non-scaled’ skin of birds and the ventral scales of crocodylians. This seems like an odd comparison, given that there is a well-established relationship in birds, which have thinner stratum corneum in non-feathered regions, whereas it is thicker on the scaled legs (see various Stettenheim references for e.g.). Similarly, since the current paper

relates to scales of *Psittacosaurus*, it would seem more appropriate to comment on scales in birds rather than the non-scaled regions. Further consideration will then also need to be made considering that avian scales are non-homologous with reptilian scales. Subsequently, it feels like an ‘apples to oranges’ comparison at present. Secondly, while the comparisons are topographically equivalent between crocs and *Psittacosaurus* (i.e. ventral scales), it would also be worth commenting on the variation between body regions in crocs. I am unsure how absolute values for stratum corneum thickness vary with ontogeny in modern reptiles and birds, but I encourage the authors to explore and comment on this potential variation as well as it has strong bearing on the current paper (i.e. a 3-6m long *C. porosus* may not be a suitable comparison with a juvenile *Psittacosaurus*!). Considering both of these aspects would greatly strengthen their argument. The conclusion on line 262 that “the original skin thickness to have fallen within the range of thicknesses exhibited by extant crocodilians and not birds” therefore needs further consideration. An additional sentence comparing your values to those reported in hadrosaurs by Fabbri et al (2019) and Barbi et al (2019) would also be relevant here. The latter paper was not cited, but should be (see also line 49).

Barbi, M., Bell, P.R., Fanti, F., Dynes, J.J., Kolaceke, A., Buttigieg, J., Coulson, I.M. and Currie, P.J., 2019. Integumentary structure and composition in an exceptionally well-preserved hadrosaur (Dinosauria: Ornithischia). *PeerJ*, 7, p.e7875.

Response: Agreed and we have revised the manuscript accordingly. Specifically, we have added anatomical comparisons of the stratum corneum of *Psittacosaurus* with that of (1) crocodilian scales (regarding variation in body regions and ontogeny; lines 304–307, 325–329), (2) avian scutate and reticulate scales (lines 307–309, 335–338) and (3) the reported hadrosaur scales (lines 294–300, 309–311). These revisions are provided in the Discussion section “Anatomy of non-feathered skin in *Psittacosaurus*”.

Comment 5: Lines 398-399: The authors state that *Psittacosaurus* “clearly exhibits the reptilian condition”, but as mentioned, there needs to be a more detailed comparison to the scales of birds before this statement can be considered.

Response: Palaeontological and developmental evidence supports the hypothesis that avian scales are secondarily derived from feathers and are thus non-homologous with reptilian scales, as also noted by the reviewer (in Comment 4). As such, avian scales are not relevant in terms of the evolutionary transition from scales to feathers. We therefore maintain that it is appropriate to compare *Psittacosaurus* skin with reptilian scales (the pleiomorphic condition) and avian feathered skin (the derived condition) when addressing the scale-feather transition. We have revised the Discussion section “Evolutionary implications” (lines 449–456) to make our reasoning clearer.

Comment 6: Lines 172-185: While I have no a priori concerns about the interpretation of microbodies, the figures do not do them justice: I cannot see the ‘spheroidal’ nature of any of them; rather, they show up as hazy black specks in both the figures and supp info. Since these images are the only ‘evidence’, some higher res images are strongly recommended.

Response: We thank the reviewer for the suggestion of additional higher-resolution images of the mouldic melanosomes. We have attempted TEM sectioning on multiple fossil skin samples,

but unfortunately the preservation in silica does not facilitate the preparation of high-quality TEM sections as the fossil skin is too friable. We think however that the presented images are sufficient to support our conclusion, because (1) the interpretation of the microbodies as melanosomes is not disputed by any of the reviewers and (2) we relied on the presence/absence, not the shape, of the fossil melanosomes to infer colour patterns.

Comment 7: Line 209: I suspect the FTIR analysis was performed in the hope of identifying organic traces. If so, It might be worth mentioning that no peaks could be attributed to stretch vibrations for organic material, which I actually find a little surprising given the apparent fidelity of the skin layers! Something else to discuss perhaps, from a taphonomic perspective, although I understand space is tight in this format. Micron-scale preservation of features seems to be fairly regularly reported for silicified invertebrate fossils, but not, as far as I'm aware, in vertebrates.

Response: We did indeed test the fossil skin for evidence of organic compounds using FTIR analysis. Given the mode of preservation of the fossil skin (3D authigenic replacement in silica), however, the likelihood of recovering organic moieties is low (Briggs 2003). Nevertheless we have added a statement regarding the absence of organic peaks in the Results section (lines 259–260) of the revised manuscript.

Reference

Briggs, D.E. The role of decay and mineralization in the preservation of soft-bodied fossils. *Annu. Rev. Earth Planet. Sci.* 31, 275–301 (2003).

Comment 8: Lines 341-451: The authors state that the specimen had not undergone “bloat and float” based on the articulation of element and the intact mass of gastroliths. While I agree that the specimen did not remain exposed and likely hadn't been dead for long, it is difficult to imagine how a terrestrial dinosaur could end up within a fine-grained, presumably deep lacustrine environment had it not floated out to its final resting place. A bloat and float scenario would seem a better explanation for why we have an animal (1) preserved belly-up (2) in a quiet, deep water setting. While I agree the degree of articulation may appear at odds with the findings of, say, Syme and Salisbury (2014), the other evidence is more compelling in this case. Perhaps of interest, the overall proportions of a juvenile *Psittacosaurus* (especially this one) do recall the proportions of ankylosaurs, which are frequently overturned, presumably also as a result of bloat and float (see Mallon et al. 2018 *Pal. Pal. Pal.*). One other question pertinent to this argument is: is the specimen actually preserved belly up, or has it just been prepared that way? Although the provenance of the specimen is unknown, there should (hopefully) be some sedimentary markers that indicate way up. This did not come across clearly in the results, which simply state “It preserves a near-complete, well-articulated skeleton in ventral aspect”. The orientation should be backed-up and confirmed as it is relevant to the taphonomy of the specimen. For the record, I have no issue with the other aspects of their taphonomic interpretation.

Response: The specimen is indeed preserved ventral surface uppermost based on soft-sediment deformation structures caused by the impacting carcass at deposition and normal

grading in the matrix sediment – we have revised the Results section (lines 79–80) and provided additional images of the sedimentary structures (Supplementary Fig. 1) to make this clear.

We are not entirely certain in what sense the reviewer refers to “bloat and float”. In much of the taphonomic literature, this phrase refers to the (partial) refloating of carcasses after deposition due to the build-up of decay gases. However the reviewer seems to use a different meaning, i.e. the floating of a carcass during transport and prior to deposition. In either case, we feel that any period of extended flotation at or near the water surface likely would have resulted in much more extensive disarticulation than observed in NJUES-10. Further, many (if not most) of the terrestrial vertebrates from the Jehol Biota are well-articulated and preserved in fine-grained lacustrine deposits (Xu et al. 2020). This contrasts with the extensive disarticulation of the carcasses frequently associated with the “bloat and float” stage in taphonomic experiments (as noted also by the reviewer).

Therefore, we feel it is more likely that carcasses of terrestrial animals were transported to the central parts of the Jehol lake(s) via surface currents arising from fluvial input and did not go through a “bloat and float” stage. The belly-up posture of NJUES-10 could have been accidental: the overall rounded shape of the body in vertical section would have rendered the animal unstable in the water column. Indeed, *Psittacosaurus* preserved with the dorsal side up is also known from the Jehol Biota (e.g. Jiang et al. 2014).

References

Jiang, B., Harlow, G.E., Wohletz, K., Zhou, Z. & Meng, J. New evidence suggests pyroclastic flows are responsible for the remarkable preservation of the Jehol biota. *Nat. Commun.* 5, 3151 (2014).

Xu, X., Zhou, Z., Wang, Y. and Wang, M. Study on the Jehol Biota: recent advances and future prospects. *Sci. China Earth Sci.* 63, 757–773 (2020).

I hope these comments are helpful.

Regards,

Phil Bell

Comments in the attached document:

Comment 9: (Line 30) Highly contentious. I suggest modifying: "possibly even dinosaurs".

Response: Agreed. We have revised the Introduction (lines 39–47) to acknowledge the controversy regarding feather origins.

Comment 10: (Line 49) Consider adding Barbi et al 2019, who also describe the stratum corneum in a hadrosaur:

Barbi, M., Bell, P.R., Fanti, F., Dynes, J.J., Kolaceke, A., Buttigieg, J., Coulson, I.M. and Currie, P.J., 2019. Integumentary structure and composition in an exceptionally well-preserved hadrosaur (Dinosauria: Ornithischia). PeerJ, 7, p.e7875.

Response: Agreed and we have revised accordingly (line 64).

Comment 11: (Line 76) This seems fairly straightforward from your description, but could you please explicitly state whether there is any variation in this pattern on different parts of the body?

Response: Agreed. We have provided additional description of the scalation in the Results section (lines 93–100) and Supplementary Information (section “Variation in preserved scale morphology in NJUES-10” and Supplementary Fig. 3).

Comment 12: (Line 104, Fig. 2) Scale bar in (h) overlaps the region on interest. Please move the scale bar so it is visible.

Response: Agreed and we have revised Fig.2 accordingly.

Comment 13: (Line 132) does this refer to the regions between vertical fractures e.g. dashed line in Fig 2i. Please clarify.

Response: Yes, the sublayer fragments refer to the regions between vertical fractures. This was clarified in the original manuscript (line 114; line 138 in the revised manuscript); we have added references to specific figures in the revised manuscript (lines 157–158).

Comment 14: (Line 137) scutate?

Response: Yes, the dorsal scales refer to the scutate scales; we have revised accordingly (line 162).

Comment 15: (Line 219, Fig. 4) It would be helpful to mark the skin-sediment boundary in the SEM image.

Response: Agreed and we have revised Fig. 4 accordingly.

Comment 16: (Line 285) Perhaps define this acronym again here since the definition only appears on pg 1.

Response: Agreed and we have revised accordingly (line 333).

Comment 17: (Line 288) what about avian scales?

Response: We have provided the relevant discussion in the revised manuscript (line 335–338).

Comment 18: (Line 398) Again, avian scales need to be more closely considered here.

Response: As in our response to Comment 5, avian scales are not relevant in terms of the evolutionary transition from scales to feathers, given that avian scales are secondarily derived from feathers and non-homologous with reptilian scales. Instead, it is appropriate to compare *Psittacosaurus* skin with reptilian scales (the pleiomorphic condition) and avian feathered skin (the derived condition) when addressing the scale-feather transition. We have revised the Discussion section “Evolutionary implications” (lines 449–456) to make our reasoning clearer.

Comment 19: (Line 427) You should differentiate between types of avian skin, since avian scales are secondarily derived structures (i.e. derived).

Response: This comment is effectively the same as Comment 18. Please see our response to Comment 18.

Comment 20: (Line 459) Presumably these are the three locations identified in Fig 1. Please reference the figure components here.

Response: Agreed and we have revised accordingly (lines 520–522).

Reviewers' Comments:

Reviewer #1:

Remarks to the Author:

I am glad to see that the authors have considered all my remarks and suggestions and would, therefore, suggest to accept the revised version of their manuscript for publication. I admittedly did not know about the lack of correlation between melanosome geometry and colour in reptilian scales and I appreciate the fact that this was discussed by the authors in the Supplementary Information. I have no other remarks to provide at this stage and only feel the need to apologize for the numerous English mistakes I left in my review.

Sincerely,

Christophe Hendrickx, San Miguel de Tucumán, the 14th of March 2024

Reviewer #2:

Remarks to the Author:

I thank the authors for their considered responses and careful revision of the MS. I have no further comments and look forward to seeing the final published version soon.

Response to reviewers

We sincerely thank all the reviewers for their insightful and constructive comments and suggestions on our manuscript. We are glad that our revisions and responses from last submission are satisfactory to the reviewers.

REVIEWER COMMENTS

Reviewer #1 (Remarks to the Author):

I am glad to see that the authors have considered all my remarks and suggestions and would, therefore, suggest to accept the revised version of their manuscript for publication. I admittedly did not know about the lack of correlation between melanosome geometry and colour in reptilian scales and I appreciate the fact that this was discussed by the authors in the Supplementary Information. I have no other remarks to provide at this stage and only feel the need to apologize for the numerous English mistakes I left in my review.

Sincerely,

Christophe Hendrickx, San Miguel de Tucumán, the 14th of March 2024

Reviewer #2 (Remarks to the Author):

I thank the authors for their considered responses and careful revision of the MS. I have no further comments and look forward to seeing the final published version soon.